# Protection against symptomatic dengue infection by neutralizing antibodies varies by infection history and infecting serotype

Sandra Bos [1,9] ✉, Aaron L. Graber[1,9], Jaime A. Cardona-Ospina[1,2], Elias M. Duarte[1], Jose Victor Zambrana[3,4], Jorge A. Ruíz Salinas [3], Reinaldo Mercado-Hernandez[1], Tulika Singh[1], Leah C. Katzelnick[5], Aravinda de Silva [6], Guillermina Kuan[3,7], Angel Balmaseda[3,8] & Eva Harris [1] ✉

Dengue viruses (DENV1–4) are the most prevalent arboviruses in humans and a major public health concern. Understanding immune mechanisms that modulate DENV infection outcome is critical for vaccine development. Neutralizing antibodies (nAbs) are an essential component of the protective immune response, yet their measurement often relies on a single cellular substrate and partially mature virions, which does not capture the full breadth of neutralizing activity and may lead to biased estimations of nAb potency. Here, we analyze 125 samples collected after one or more DENV infections but prior to subsequent symptomatic or inapparent DENV1, DENV2, or DENV3 infections from a long-standing pediatric cohort study in Nicaragua. By assessing nAb responses using Vero cells with or without DC-SIGN and with mature or partially mature virions, we find that nAb potency and the protective NT50 cutoff are greatly influenced by cell substrate and virion maturation state. Additionally, the correlation between nAb titer and protection from disease depends on prior infection history and infecting serotype. Finally, we uncover variations in nAb composition that contribute to protection from symptomatic infection differently after primary and secondary prior infection. These findings have important implications for identifying antibody correlates of protection for vaccines and natural infections.

The four dengue virus serotypes (DENV1-4) are mosquito-borne flaviviruses endemic to tropical and subtropical regions that cause the most prevalent arthropod-borne viral disease in humans. Up to 100 million people are infected each year, of whom approximately 50 million develop symptoms ranging from self-limiting but debilitating dengue fever to potentially fatal dengue shock syndrome[1]. Infection with one of the four antigenically related DENV serotypes leads to the production of type-specific antibodies, which are specific to the infecting serotype, as well as cross-reactive antibodies that can bind to other DENV serotypes. These cross-reactive antibodies can either

[1]Division of Infectious Diseases and Vaccinology, School of Public Health, University of California, Berkeley, Berkeley, CA, USA. [2]Grupo de Investigación Biomedicina, Facultad de Medicina, Institución Universitaria Visión de las Américas, Pereira, Colombia. [3]Sustainable Sciences Institute, Managua, Nicaragua. [4]Department of Epidemiology, School of Public Health, University of Michigan, Ann Arbor, MI, USA. [5]Viral Epidemiology and Immunity Unit, Laboratory of Infectious Diseases, National Institute of Allergy and Infectious Diseases, National Institutes of Health, Bethesda, MD, USA. [6]Department of Microbiology and Immunology, University of North Carolina, Chapel Hill, NC, USA. [7]Centro de Salud Sócrates Flores Vivas, Ministerio de Salud, Managua, Nicaragua. [8]Laboratorio Nacional de Virología, Centro Nacional de Diagnóstico y Referencia, Ministerio de Salud, Managua, Nicaragua. [9]These authors contributed equally: Sandra Bos, Aaron L. Graber. ✉e-mail: sbos@berkeley.edu; eharris@berkeley.edu

provide protection or increase the risk of subsequent symptomatic or severe DENV infection after both natural infections and vaccines[2–6]. The occurrence of such adverse events emphasizes the complexity of DENV immunity and the crucial need for a comprehensive evaluation of correlates of protection to facilitate studies of natural infections and the development of safe and effective DENV vaccines.

Neutralizing antibodies (nAbs) are essential for protection against dengue in both natural infections and vaccine trials[5,7]. The composition and magnitude of pre-existing nAbs can vary depending on the previous infecting serotype and number of previous infections[2]. The current "gold standard" to assess the neutralizing potential of nAbs are Plaque or Focus Reduction Neutralization Tests (PRNT/FRNT), where serum/plasma samples are serially diluted to measure viral infectivity[8]. Vaccine trials to evaluate correlates of protection usually perform DENV neutralization assays according to the reference assay conditions with Vero cells as substrate and WHO reference DENV strains grown in either mammalian (Vero) or insect (C6/36) cells[8]. Although higher nAb titers correlate with a reduced risk of dengue disease, symptomatic dengue cases can occur in individuals with relatively high nAb titers, suggesting that the current neutralization assays may not adequately measure the types of nAbs critical for protection[5,7].

The DENV genome encodes three structural proteins -- envelope (E), precursor membrane/membrane (prM/M), and capsid (C) -- that form the virion and seven nonstructural proteins responsible for viral replication, assembly, and immune evasion[9]. Neutralizing antibodies primarily target the envelope (E) protein of the virus, and nAb potency is influenced by the epitope they target. The epitope itself determines the amount of antibodies required to neutralize a viral particle (stoichiometry), and their efficiency varies depending on viral strain and physiological environment. Thus, several factors, including the temperature of incubation, the maturation state of the virion, the viral strain, and the choice of cell substrate impact the type of antibody measured. As a result, neutralization assay conditions can modulate nAb potency[10,11] and the ability to measure antibody "repertoire" composition and functionality, referred to here as "quality".

DENV FRNT/PRNTs are commonly performed using DENV laboratory strains grown in Vero cells, which may not reflect the maturation state of virus circulating in human participants[10]. During viral assembly, immature particles are formed in the endoplasmic reticulum and transported to the extracellular surface through the Golgi network, where they undergo maturation. These immature particles are characterized by a spiky surface formed by trimeric prM-E heterodimers, with pr peptides capping the E fusion loop (FL)[12]. Their transit through the acidic compartment of the Golgi network results in a rearrangement of the E proteins (trimers to dimers), and maturation occurs by cleavage of prM by the host protease furin[13,14]. Upon release into the neutral pH of the extracellular medium, the pr peptides detach, revealing the smooth mature viral progeny that are infectious. However, maturation is not always complete. Areas where pr has not been cleaved revert to trimeric spikes once virions are secreted, creating mosaic particles harboring different organizations of E proteins on their surface[15]. This difference in maturation state changes the proportion and identity of epitopes exposed and, thus, the number and type of antibodies required to completely neutralize infectious viral particles[11,16].

In addition, the cellular substrate of the neutralization assay is an important factor that can influence neutralization potency, as it determines the host receptors relevant for viral entry. Unlike other flaviviruses, the E protein of DENV has an additional glycosylation site at Asp-67 (E domain I), which allows DENV to enter cells through interaction with the DC-SIGN attachment factor[17]. DC-SIGN is expressed on dendritic cells (DCs) and monocytes, which account for the majority of DENV-infected cells in the blood[18–23]. DCs are potent antigen-presenting cells that mediate both innate and adaptive immunity by producing cytokines and transporting antigens to

secondary lymphoid organs, where they prime naïve antigen-specific T cells. Nevertheless, in vitro studies have shown that while DENV2-infected DCs can undergo activation and release of pro-inflammatory cytokines, they do not secrete IFN-α/β, which is critical for T cell priming[20]. Meanwhile, monocytes facilitate viral dissemination to various peripheral organs by acting as Trojan horses and transporting DENV virions to target tissues[24]. DC-SIGN promoter polymorphism in humans contributes to genetic susceptibility to DENV infection and is a risk factor for developing severe dengue disease. As a result, efficient neutralization in the presence of DC-SIGN is important for limiting viral dissemination and for the establishment of a rapid and effective cell-mediated immune response[20]. Evaluation of DC-SIGN-mediated entry in an FRNT/PRNT assay enables a broad assessment of neutralizing potential of nAbs by measuring the impact of nAbs that prevent N-glycan interaction and should be considered when investigating immune correlates of protection.

Here, we used a set of plasma samples from our long-standing prospective pediatric dengue cohort study in Nicaragua and evaluated the cross-reactive nAb response prior to subsequent inapparent or symptomatic infections with DENV1, DENV2, or DENV3. We compared neutralization potency across multiple assay conditions using partially mature and mature virions on Vero cells and Vero cells expressing DC-SIGN. Using these four assay conditions, we evaluated the nAb magnitude and quality associated with protection against symptomatic DENV infection. We found that correlation with protection is dependent on the individual's prior DENV infection history and the subsequent infecting serotype, as well as assay conditions. Further, we show that comparison of nAb titers obtained with partially mature versus mature virions and presence or absence of DC-SIGN can indicate the characteristics of antibodies that are important for protection.

## Results

### Participants and DENV infections

In this study, we analyzed pre-infection samples from 125 participants in the PDCS in Managua, Nicaragua[6,25]. These samples were collected 6–12 months before an inapparent DENV infection (pre-inapparent group, $n = 63$) or before a symptomatic infection (pre-symptomatic group, $n = 62$) (Fig. 1). All participants included in the study had previously experienced DENV infection and had developed anti-DENV antibodies (i.e., DENV-immune).

### Characterization of virus maturation state and infectivity

To evaluate how the neutralization potency of serum antibodies is affected by DENV maturation state, we first generated partially mature and mature virion stocks using clinical isolates of DENV1, DENV2, and DENV3 from Nicaragua. The maturation state of these virions was confirmed by analyzing the prM content in each virus stock via Western blot analysis. We observed significant prM levels in viruses grown in standard Vero cells, indicative of a partially mature phenotype (Supplementary Fig. S1A). In contrast, production of the same parental viruses in furin-overexpressing cells led to a substantial increase in

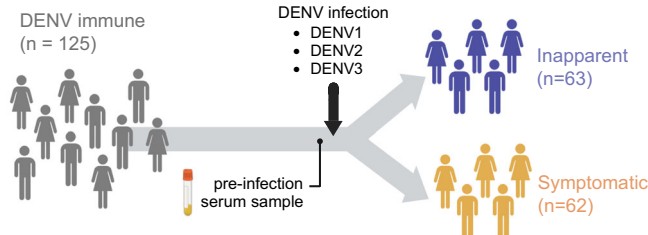

**Fig. 1 | Participant serological sampling schematic.** Plasma samples from 125 DENV-immune participants were collected before a subsequent DENV1, 2, or 3 infection and were stratified into two separate groups according to disease outcome: Pre-inapparent (purple, $n = 63$) and pre-symptomatic (yellow, $n = 62$).

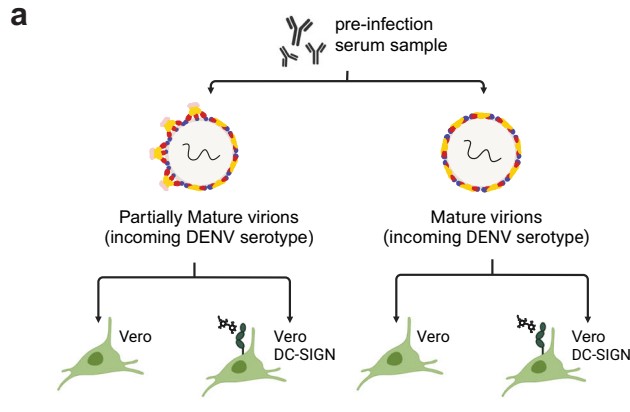

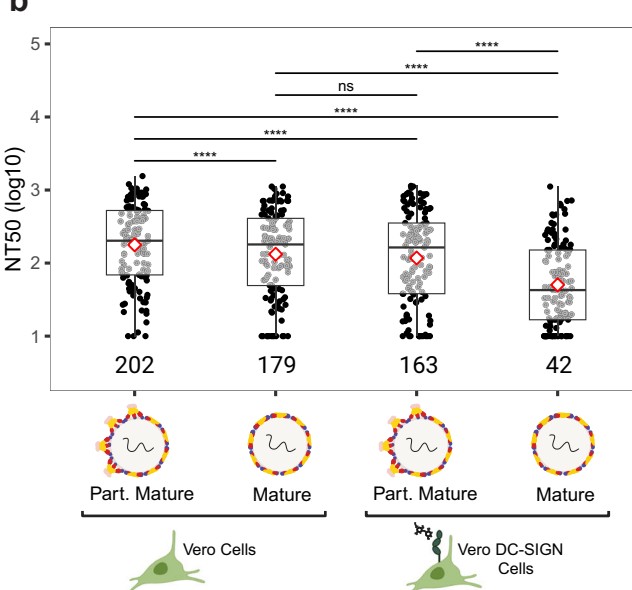

**Fig. 2 | Cross-reactive nAb titers across assay conditions. a** Schematic of the different neutralization assay conditions. **b** Neutralizing titers ($NT_{50}$) of 125 participants measured against the incoming DENV serotype using virions of different maturation state and different cell substrate. Median $NT_{50}$ values are indicated below each boxplot. Shown are median $NT_{50}$ (middle line), 25th to 75th percentile (box), and 5th to 95th percentile (whiskers) as well as the raw data (points). Diamond indicates mean $NT_{50}$. Asterisks indicate Benjamini–Hochberg adjusted p-values for paired Wilcoxon test. p-values: ns >0.05; *<0.05; **<0.01; ***<0.001; ****<0.0001. Part. Mature partially mature. Source Data are provided as a Source Data file.

entry through interaction with the N67 glycan on the envelope protein[17]. Viral infectivity was quantified by determining the ratio of physical particle (vRNA) equivalents to focus forming units (FFU) (Supplementary Fig. S1B). As anticipated, the presence of DC-SIGN on the surface of Vero cells enhanced the infectivity of both partially mature and mature DENV stocks, indicating that DC-SIGN provides an alternative and potentially more favorable entry pathway for DENV. Importantly, the lowest infectivity ($10^3$–$10^4$ vRNA:FFU) was observed using partially mature virions with standard Vero cells, corresponding to the conditions of the reference neutralization assay. In contrast, the highest infectivity ($10^2$–$10^3$ vRNA:FFU) was observed using mature virions with Vero DC-SIGN cells (Supplementary Fig. S1B). These results show that under reference assay conditions, DENV virions exhibit infectivity levels approximately ten times lower compared to conditions that more closely resemble the DENV maturation state and entry pathway in vivo.

## Cell substrate and DENV maturation state impact neutralization titer

Given the substantial influence of maturation state and cellular substrate/entry pathway on DENV infectivity, we next wanted to evaluate how these parameters impact the measurement of neutralizing antibody titers ($NT_{50}$), which are often used as a surrogate for protection against dengue disease. To address this question, we used pre-infection samples from the 125 DENV-immune participants included in this study, collected prior to a subsequent DENV infection, and evaluated how the neutralization potency of antibodies from each participant varied across four assay conditions: partially mature DENV with Vero cells, partially mature DENV with Vero DC-SIGN cells, mature DENV with Vero cells, and mature DENV with Vero DC-SIGN cells (Fig. 2a). In each case, $NT_{50}$ titers were measured against the heterotypic incoming serotype of each participant and compared using a paired Wilcoxon test. For instance, the $NT_{50}$ of serum samples from participants collected prior to a DENV1 infection was measured using DENV1, while the $NT_{50}$ of serum samples from participants collected prior to a DENV2 infection was measured using DENV2, etc.

Our results demonstrate that overall, $NT_{50}$ values were significantly lower in assays performed with mature virions compared to partially mature virions, indicating that serum antibodies had a lower potency for neutralizing the incoming DENV virion in its mature form. Additionally, $NT_{50}$ values were lower in assays performed on Vero DC-SIGN cells compared to Vero cells (Fig. 2b), indicating a decrease in the potency of nAbs under conditions that facilitate glycan-mediated DENV entry. Notably, the most significant difference in median $NT_{50}$ values was observed between assays performed using Vero cells infected with partially mature virions (reference assay condition, $NT_{50}$ = 202) and those using Vero DC-SIGN cells infected with mature virions ($NT_{50}$ = 42).

These findings underscore the significant impact of virion maturation state and cell substrate on the measurement of nAbs titers. Furthermore, they suggest that under reference assay conditions, the potency of cross-reactive nAbs from the same participant may be overestimated compared to neutralization assays performed with more mature virions and in the presence of DC-SIGN.

## High pre-existing cross-reactive neutralizing antibody titer is associated with protection from dengue disease

We next investigated whether pre-infection nAb titers are associated with protection from dengue disease. We compared the $NT_{50}$ of pre-inapparent (n = 64) versus pre-symptomatic (n = 63) DENV infection samples using the four assay conditions described previously. Our observations revealed a significantly higher $NT_{50}$ in the pre-inapparent group, composed of individuals who were protected against dengue disease, as compared to the pre-symptomatic group across all four

maturation, as evidenced by low to undetectable levels of prM (Supplementary Fig. S1). To further validate virion maturation state, we performed FRNT using anti-FL mAbs (1M7, 4G2), which neutralize partially mature virions better than mature virions[10]. As expected, virions grown in Vero-furin cells exhibited greater resistance to neutralization by anti-FL mAbs, requiring higher concentrations for neutralization compared to virions grown in standard Vero cells (Supplementary Fig. S1C, D). These results demonstrated that viruses grown in furin-overexpressing Vero cells were more mature than those grown in standard Vero cells. Henceforth, we will refer to viruses grown in standard and furin-overexpressing Vero cells as partially mature and mature virions, respectively.

We then evaluated the infectivity of both partially mature and mature DENV1, DENV2, and DENV3 on standard Vero cells and Vero cells expressing the cellular attachment factor DC-SIGN. This C-type lectin receptor plays a pivotal role in DENV biology by facilitating cell

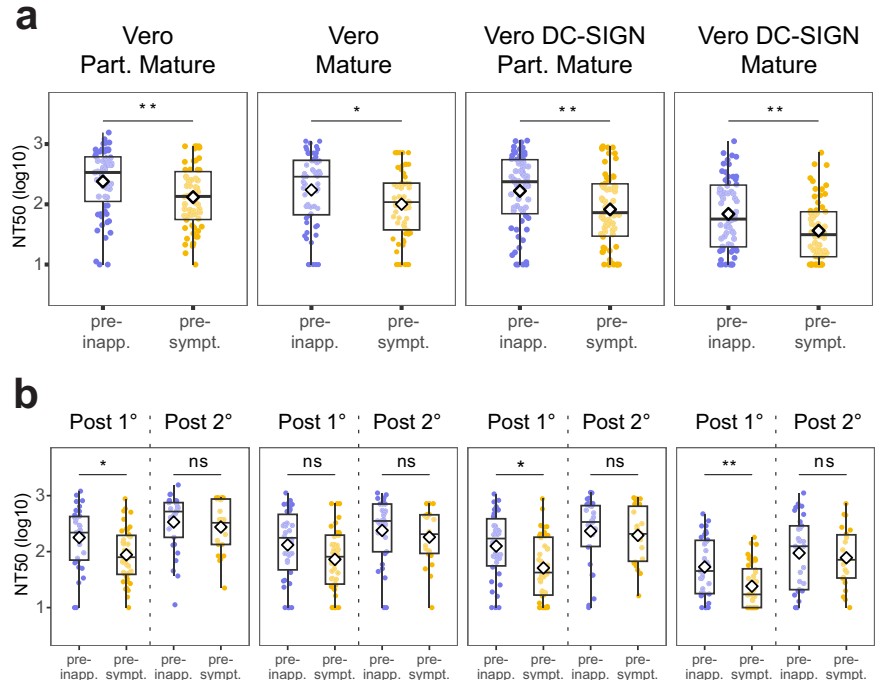

**Fig. 3 | Cross-reactive nAb titers of pre-inapparent and pre-symptomatic DENV infection samples. a** $NT_{50}$ of pre-inapparent (purple, $n = 63$) and pre-symptomatic (yellow, $n = 62$) DENV infection groups measured against the incoming serotype across FRNT assay conditions, and **b** $NT_{50}$ of the same pre-inapparent and pre-symptomatic DENV infection group stratified by participants' infection history at the time of sample collection. Shown are median $NT_{50}$ (middle line), 25th to 75th percentile (box), and 5th to 95th percentile (whiskers) as well as the raw data (points). Diamond indicates mean $NT_{50}$. Asterisks indicate Benjamini–Hochberg adjusted $p$-values for Wilcoxon test (two-sided). $p$-values: ns >0.05; *<0.05; **<0.01; ***<0.001; ****<0.0001. Post-1° post-primary infection, Post-2° post-secondary infection, pre-inapp. pre-inapparent DENV infection group (purple), pre-sympt. pre-symptomatic DENV infection group (yellow). Source Data are provided as a Source Data file.

**Table 1 | ROC analysis and logistic regression of $NT_{50}$ magnitude as a predictor for developing a subsequent symptomatic disease, stratified by infection history**

| | AUC[a] | CI 95[b] | Threshold ($NT_{50}$) | Sensitivity | Specificity | OR[c] | CI 95 |
|---|---|---|---|---|---|---|---|
| **Combined** | | | | | | | |
| Part.Mature[d] Vero | 0.64 | 0.54–0.74 | 898.13 | 0.94 | 0.08 | 0.86 | 0.75–0.96 |
| Mature Vero | 0.62 | 0.52–0.72 | 475.83 | 0.9 | 0.32 | 0.82 | 0.70–0.94 |
| Part.Mature[d] Vero DC-SIGN | 0.65 | 0.55–0.75 | 701.95 | 0.9 | 0.14 | 0.85 | 0.73–0.96 |
| Mature Vero DC-SIGN | 0.64 | 0.54–0.74 | 180.25 | 0.9 | 0.29 | 0.72 | 0.53–0.92 |
| **Primary** | | | | | | | |
| Part.Mature[d] Vero | 0.68 | 0.56–0.81 | 392.70 | 0.9 | 0.29 | 0.75 | 0.58–0.93 |
| Mature Vero | 0.63 | 0.49–0.76 | 367.95 | 0.9 | 0.26 | 0.79 | 0.63–0.97 |
| Part.Mature[d] Vero DC-SIGN | 0.68 | 0.56–0.81 | 276.60 | 0.9 | 0.32 | 0.73 | 0.55–0.92 |
| Mature Vero DC-SIGN | 0.70 | 0.58–0.82 | 96.81 | 0.93 | 0.35 | 0.31 | 0.12–0.65 |
| **Secondary** | | | | | | | |
| Part.Mature[d] Vero | 0.54 | 0.37–0.71 | 981.84 | 1 | 0.10 | *ns* | – |
| Mature Vero | 0.59 | 0.43–0.75 | 767.76 | 1 | 0.17 | *ns* | – |
| Part.Mature[d] Vero DC-SIGN | 0.56 | 0.39–0.73 | 891.09 | 0.95 | 0.10 | *ns* | – |
| Mature Vero DC-SIGN | 0.54 | 0.38–0.70 | 455.15 | 0.91 | 0.21 | *ns* | – |

[a]*AUC* Area under the curve, [b]*CI95* 95% Confidence interval, [c]*OR* Odds ratio, [d]*Part. Mature* Partially mature virion.

conditions, including the reference assay condition (Fig. 3a). This finding aligns with our previous studies[7] and supports that, overall, higher nAb titers are associated with protection from symptomatic dengue.

To further analyze whether the $NT_{50}$ magnitude was associated with lower risk of a subsequent symptomatic DENV infection, we performed a logistic regression model for each assay condition. Consistently, a higher $NT_{50}$ was associated with lower risk of developing a subsequent symptomatic disease when the assay was performed using partially mature virions (OR 0.86, 95% Confidence Interval [CI] 0.75–0.96) or mature virions (OR 0.82, 95% CI 0.70–0.94) on Vero cells, as well as for partially mature virions (OR 0.85, 95% CI 0.73–0.96) and mature virions (OR 0.72, 95% CI 0.53–0.92) on Vero DC-SIGN cells (Table 1). Taken together, these results demonstrate that higher pre-existing cross-reactive nAb titers are correlated with lower risk of developing a subsequent symptomatic infection.

**Table 2 | Pre-inapparent and pre-symptomatic infection participant characteristics**

| | Pre-inapparent | Pre-symptomatic | p |
|---|---|---|---|
| n | 63 | 62 | |
| Age (mean (SD)) | 10.47 (2.86) | 10.19 (3.59) | 0.633 |
| Male (%) | 35 (55.6) | 38 (61.3) | 0.639 |
| Infection history (%) | | | 0.3 |
| Primary (%) | 34 (54.0) | 40 (64.5) | |
| Secondary (%) | 29 (46.0) | 22 (35.5) | |

Participants characteristics were compared using ANOVA for numerical or chi-square for categorical variable (two sided). No adjustments nor multiple comparison were performed.

## High cross-reactive neutralizing antibody titer is associated with protection from dengue disease in participants who have experienced a single prior DENV infection, but not in those who have had two or more DENV infections

Next, we interrogated the impact of prior immunity on protection from subsequent dengue disease by stratifying participants by their infection history. At the time of sample collection, 74/125 participants had experienced a single DENV infection (primary infection), and 51/125 participants had experienced two or more DENV infections (secondary infection). In the pre-inapparent group, 34/63 individuals had previously experienced a primary infection, while 29/63 had previously had secondary infection(s). In the pre-symptomatic group, 40/62 individuals had experienced a previous primary infection and 22/62 had had secondary infection(s) (Table 2). Samples collected from participants with prior primary infection will be referred to as "post-primary" (Post-1°), while samples collected from participants who previously had had two or more DENV infections will be referred to as "post-secondary" (Post-2°).

In participants with a prior primary DENV infection, we observed significantly higher $NT_{50}$ values in the pre-inapparent group compared to the pre-symptomatic group across all four conditions (Fig. 3b). This suggests that high titers of cross-reactive nAbs are associated with protection from disease upon a second DENV infection. However, in the post-secondary group, both pre-inapparent and pre-symptomatic participants exhibited high $NT_{50}$ titers, with no significant difference in titer magnitude (Fig. 3b). Notably, we observed that the titers in participants from the post-secondary pre-symptomatic infection group were as high as the titers observed in participants with one prior infection who were protected against a subsequent symptomatic infection (Post-1° pre-inapparent group) (Fig. 4b). This shows that high nAb titers alone are not sufficient for conferring protection, implying that other aspects of the nAb response beyond magnitude, or even beyond neutralization potency, may play a role in protection after secondary DENV infection.

Consistently, in the post-primary infection group, the $NT_{50}$ assessed using Vero cells with partially mature virions (OR 0.75, 95% CI 0.58–0.93) and mature virions (OR 0.79, 95% CI 0.63–0.97), as well as the $NT_{50}$ assessed using Vero DC-SIGN cells with partially mature virions (OR 0.73, 95% CI 0.55–0.92) and mature virions (OR 0.31, 95% CI 0.12–0.65), were associated with lower risk of subsequent symptomatic DENV infection using our logistic regression model (Table 1). However, in the post-secondary infection group, none of the $NT_{50}$ titers assessed were associated with protection.

Collectively, these findings highlight that the association between nAb titer magnitude and protection against symptomatic DENV infection depends on the infection history of participants. Moreover, they suggest that while robust production of cross-reactive nAbs (magnitude) is a critical factor in protection against a second DENV infection, it appears insufficient to confer protection against subsequent infections. This implies that other nuanced aspects within or beyond the nAb response might contribute more to protection in individuals with secondary DENV infection histories.

## Maturation state- and DC-SIGN-sensitive antibodies are potential drivers of symptomatic infection outcome after secondary infection

In light of these observations, we leveraged the different assay conditions to gain insight into the composition of the nAb response. To do so, we compared the neutralizing potency of nAbs from each participant across paired assay conditions. Specifically, comparison of the $NT_{50}$ measured using partially mature versus mature virions was used to evaluate the contribution of antibodies that are maturation state-dependent, while comparison of $NT_{50}$ measured on standard Vero versus Vero-DC-SIGN cells was used as a proxy to evaluate whether neutralization was mediated by antibodies capable of neutralizing DENV through N-glycan-mediated entry.

As previously mentioned, we did not observe any significant difference in the magnitude of $NT_{50}$ from post-primary pre-inapparent infection individuals (i.e., protected) and post-secondary pre-symptomatic infection individuals (i.e., not protected) across all four conditions tested. Consequently, we found these two groups particularly intriguing and compared them, along with post-secondary pre-symptomatic individuals, to investigate whether variations in nAb composition were associated with different outcomes (Fig. 4a, b). Among the characteristics that set the group of pre-symptomatic participants apart, we found that, unlike both pre-inapparent infection groups (Post-1° and Post-2°), the neutralization potency of post-secondary pre-symptomatic infection samples was significantly lower when using mature virions compared to partially mature virions in Vero cells (maturation-sensitive nAbs) (Fig. 4c). Similarly, lower neutralization potency was observed when using partially mature virions to infect DC-SIGN-expressing Vero cells compared to standard Vero cells in post-secondary pre-symptomatic infection samples (Fig. 4c). This underscores the variations that exist in the composition of the nAbs between protected and unprotected individuals. Specifically, serum samples from participants protected against dengue disease were characterized by cross-reactive nAbs that exhibited equal potency against both mature and partially mature virions in Vero cells, and in the presence or absence of DC-SIGN. In contrast, the nAb repertoire/composition of sera from participants who experienced symptomatic infection (Post-2° pre-symptomatic) was marked by nAbs sensitive to maturation state and the presence of DC-SIGN.

This finding supports that factors beyond the magnitude of nAb titers may play a substantial role in conferring protection against symptomatic dengue disease and emphasizes the importance of considering antibody quality, in addition to magnitude, when studying correlates of protection against dengue disease.

## Neutralizing antibody magnitude associates with protection against dengue disease in a serotype-dependent manner

We next investigated if the association of cross-reactive nAb titer with disease protection was similar when the infecting serotype was DENV1, DENV2 or DENV3. To address this question, we stratified the analysis by incoming DENV serotype. Our study included 19 individuals with subsequent post-primary DENV1 infections, 61 individuals with subsequent DENV2 infections, and 47 individuals with subsequent DENV3 infections (Table 3). In the pre-DENV1 infection group (Fig. 5a, top row), which includes only post-primary infection individuals, higher nAb titers were observed in the pre-inapparent group using neutralization assays with mature virions on standard Vero cells and partially mature virions on both standard Vero and Vero DC-SIGN cells.

In the pre-DENV2 infection group, significantly higher nAb titers were observed in the pre-inapparent individuals across all conditions (Fig. 5a, middle row) when samples were not stratified by infection history. However, after stratification (Fig. 5b), only nAb titers assessed in standard Vero and Vero DC-SIGN cells using mature virions were correlated with protection from symptomatic infection in the post-primary group (n = 29) (OR 0.62, 95% CI 0.38–0.88; OR 0.09, 95%

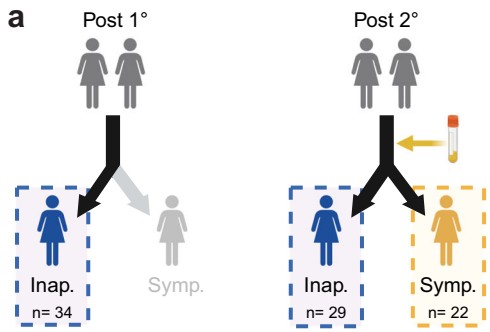

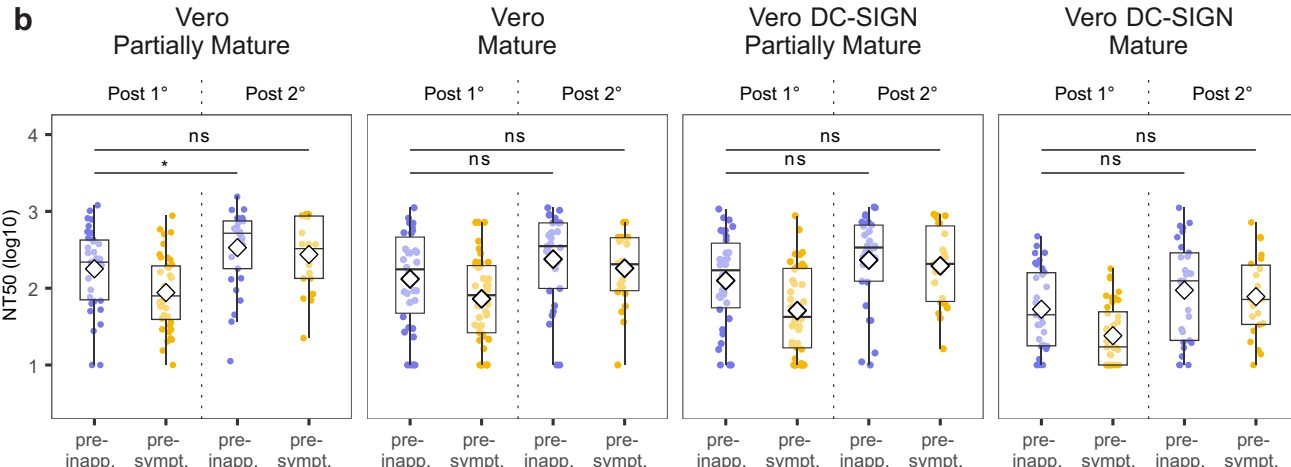

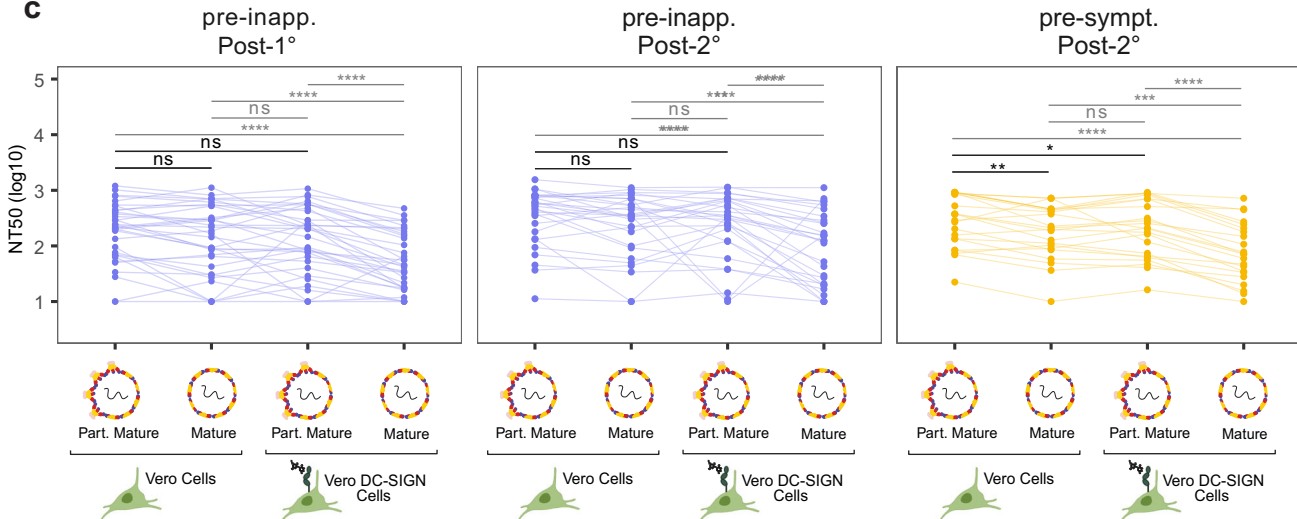

**Fig. 4 | Comparison of nAb responses between pre-inapparent post-primary and post-secondary versus pre-symptomatic post-secondary infection samples. a** Schematic of the groups compared. Sample size indicated for each group. **b** Magnitude of the nAb response using different virion maturation states and cell substrates. Shown are median $NT_{50}$ (middle line), 25th to 75th percentile (box), and 5th to 95th percentile (whiskers) as well as the raw data (points). Diamond indicates mean $NT_{50}$. **c** Impact of change in assay condition on the neutralization titers of participants. Asterisks indicate Benjamini–Hochberg adjusted $p$-values for two-sided Wilcoxon test (**b**) and paired Wilcoxon test (**c**). $p$-values: ns >0.05; *<0.05; **<0.01; ***<0.001; ****<0.0001. Post-1° post-primary infection, Post-2° post-secondary infection, pre-inapp. pre-inapparent DENV infection group (purple), pre-sympt. pre-symptomatic DENV infection group (yellow), Part. Mature partially mature. Source Data are provided as a Source Data file.

0.01–0.52, respectively) and with mature virions in standard Vero cells in the post-secondary group ($n = 32$) (OR 0.72, 95% CI, 0.5–0.95) (Fig. 5b and Supplementary Table S1).

In contrast, in the pre-DENV3 infection group (Fig. 5a, bottom row), the magnitude of nAb titers was not significantly different between subsequent symptomatic and inapparent infection regardless of assay condition or infection history (Supplementary Fig. S2).

In summary, our findings reveal serotype-specific variation in protection against dengue disease. Higher pre-infection levels of cross-reactive nAbs are associated with protection from symptomatic DENV1

**Table 3 | Participant characteristics stratified by incoming serotype and infection history**

|  | DENV1 | DENV2 | DENV3 | *p* |
|---|---|---|---|---|
| *n* | 19 | 60 | 46 |  |
| Age (mean (SD)) | 9.83 (2.08) | 11.17 (3.63) | 9.45 (2.80) | 0.018 |
| Male (%) | 11 (57.9) | 35 (58.3) | 27 (58.7) | 0.998 |
| Group |  |  |  | 0.579 |
| Pre-symptomatic (%) | 10 (52.6) | 32 (53.5) | 20 (43.5) |  |
| Pre-inapparent (%) | 9 (47.4) | 28 (46.5) | 26 (56.5) |  |
| Infection history (%) |  |  |  | <0.001 |
| Primary (%) | 19 (100.0) | 29 (48.3) | 26 (56.5) |  |
| Secondary (%) | 0 (0.0) | 31 (51.7) | 20 (43.5) |  |

Participants characteristics were compared using ANOVA for numerical or chi-square for categorical variable (two sided). No adjustments nor multiple comparison were performed.

and DENV2 infections. However, this association was not observed in the case of DENV3, suggesting that disease protection in the context of DENV3 infections might involve other aspects of the immune response.

## Pre-infection neutralizing antibody composition differentially impacts outcome by DENV serotype and infection history

To assess how variations in nAb composition in pre-infection sera relate to protection from symptomatic DENV infection for each serotype, we used paired Wilcoxon tests to compare the neutralization potency of samples stratified by serotype across assay conditions. In post-primary pre-DENV1, pre-DENV2, and pre-DENV3 symptomatic individuals (Fig. 6a–c), we observed significant differences in $NT_{50}$ titer when FRNT assays were performed using partially mature virions with standard Vero compared to Vero DC-SIGN cells. These findings indicate that antibodies present in pre-symptomatic infection samples are less potent for neutralizing DENV when cell entry is glycan-mediated (i.e., interacting with DC-SIGN on the target cell). This suggests a difference in nAb composition, with pre-symptomatic post-primary DENV infection individuals having less nAbs able to block DC-SIGN-mediated entry.

In contrast, in post-secondary pre-DENV2 infection samples (Fig. 6d), a significant reduction in $NT_{50}$ titer was observed in pre-symptomatic infection samples when FRNT assays were performed in standard Vero cells using partially mature versus mature DENV2 virions. This suggests a difference in nAb composition, with post-secondary pre-symptomatic DENV2 individuals having more nAbs sensitive to maturation state.

Interestingly, no difference in nAb quality was observed in post-secondary pre-DENV3 infections.

## Assay condition impacts cross-reactive neutralizing antibody titer cut-off to distinguish subsequent inapparent and symptomatic DENV infections

Since the identification of individuals who will develop a subsequent symptomatic infection is relevant for vaccine design, we evaluated the performance of the four assay conditions for detecting a subsequent symptomatic DENV infection. We performed a ROC analysis to compare the assay conditions in terms of their sensitivity and specificity (Table 1). The cut-off of the assay was defined as the $NT_{50}$ value under which at least 90% of the subsequent symptomatic infections were detected (sensitivity). The specificity corresponded to the proportion of inapparent infections above the cut-off value.

Notably, we found a substantial impact of assay condition on the test performance for discriminating individuals who would subsequently develop a symptomatic versus inapparent infection. The $NT_{50}$ cut-off varied substantially, ranging from 180 for mature virions on Vero DC-SIGN cells to 898 for partially mature virions on Vero cells

(Table 1), and the maximum specificity achieved was only 32% for the assay performed on mature virions with Vero cells (Table 1). For the assay performed on partially mature virions on standard Vero cells, which is the assay used in most clinical trials for vaccine testing, the $NT_{50}$ cut-off for achieving a sensitivity of at least 90% was 898.13. Nevertheless, at this titer, the specificity was only 8%. This means that, using this assay, although more than 90% of the individuals had an $NT_{50}$ below 898 before developing a symptomatic DENV infection, 92% of the individuals who subsequently developed an inapparent infection were also below this cut-off and yet were protected from disease.

When we stratified by infection history, all assay conditions effectively identified subsequent symptomatic infections in the post-primary and post-secondary infection group with at least 90% of sensitivity (Table 1). However, the specificity ranged from 29% to 35% in the post-primary infection group, and from 10% to 21% in the post-secondary infection group. Interestingly, mature virions with Vero DC-SIGN cells displayed the best performance in both groups (Table 1). This reveals that in the context of natural infection, assessment of nAb titers are appropriate screening tests for identifying individuals at risk for developing subsequent symptomatic DENV disease, but they perform poorly for detecting protected individuals with subsequent inapparent infection.

## Discussion

In this study, we analyzed 125 individuals with known infection history who experienced subsequent inapparent or symptomatic DENV infection from our long-standing cohort in Nicaragua. We evaluated the effect of previous primary or secondary DENV immunity on the quantity and quality of the cross-reactive nAb response by controlling key parameters of the neutralization assay. We also assessed the association of nAb titer under different assay conditions with probability of symptomatic DENV1, DENV2 or DENV3 infection. While we confirmed that nAb titers overall are associated with protection against symptomatic DENV infection[7], we found important differences by infection history (i.e., prior primary versus secondary immunity) and incoming serotype. We also show that the ability of nAb titers to identify individuals at risk depends on DENV maturation state and cellular substrate in the neutralization assay.

Neutralizing antibodies are considered an essential component of the protective response against flavivirus infections and are used to assess vaccine safety and efficacy[26,27]. As a quantitative functional antibody assay, the neutralization assay is considered the "gold standard" for quantifying and characterizing DENV nAbs[8]. However, in routine laboratory practice as well as in clinical trials, it is common to use virions grown in Vero or C6/36 cells[8], although it has been shown that DENV virions purified directly from human blood or produced in primary cells using monocyte-derived dendritic cells are more mature than virions grown in cell culture-adapted lines[10,28]. To approach a virion state closer to the one observed during natural infection, furin-overexpressing cell lines were generated and used to produce mature flavivirus stocks with low prM content[28,29]. This difference in maturation state changes the proportion and identity of epitopes exposed and, thus, the number and type of antibodies required to completely neutralize infectious particles[11,16]. Consistent with published literature, we observed that plasma samples showed a significantly lower neutralizing activity on mature virions compared to partially mature virions grown in Vero cells, suggesting that neutralization titers may be overestimated under standard FRNT conditions[30].

Glycan-mediated entry is an important aspect of DENV dissemination in humans[31,32], but standard assay conditions using cells lacking C-type lectin receptors (e.g., DC-SIGN) do not measure the contribution of nAbs that prevent N-glycan interactions. Unlike other flaviviruses, the E protein of DENV has an additional glycosylation site at Asp-67 (EDI), which allows DENV to enter cells through interaction with DC-SIGN[17]. As we and others have shown[17,33], DC-SIGN is a key

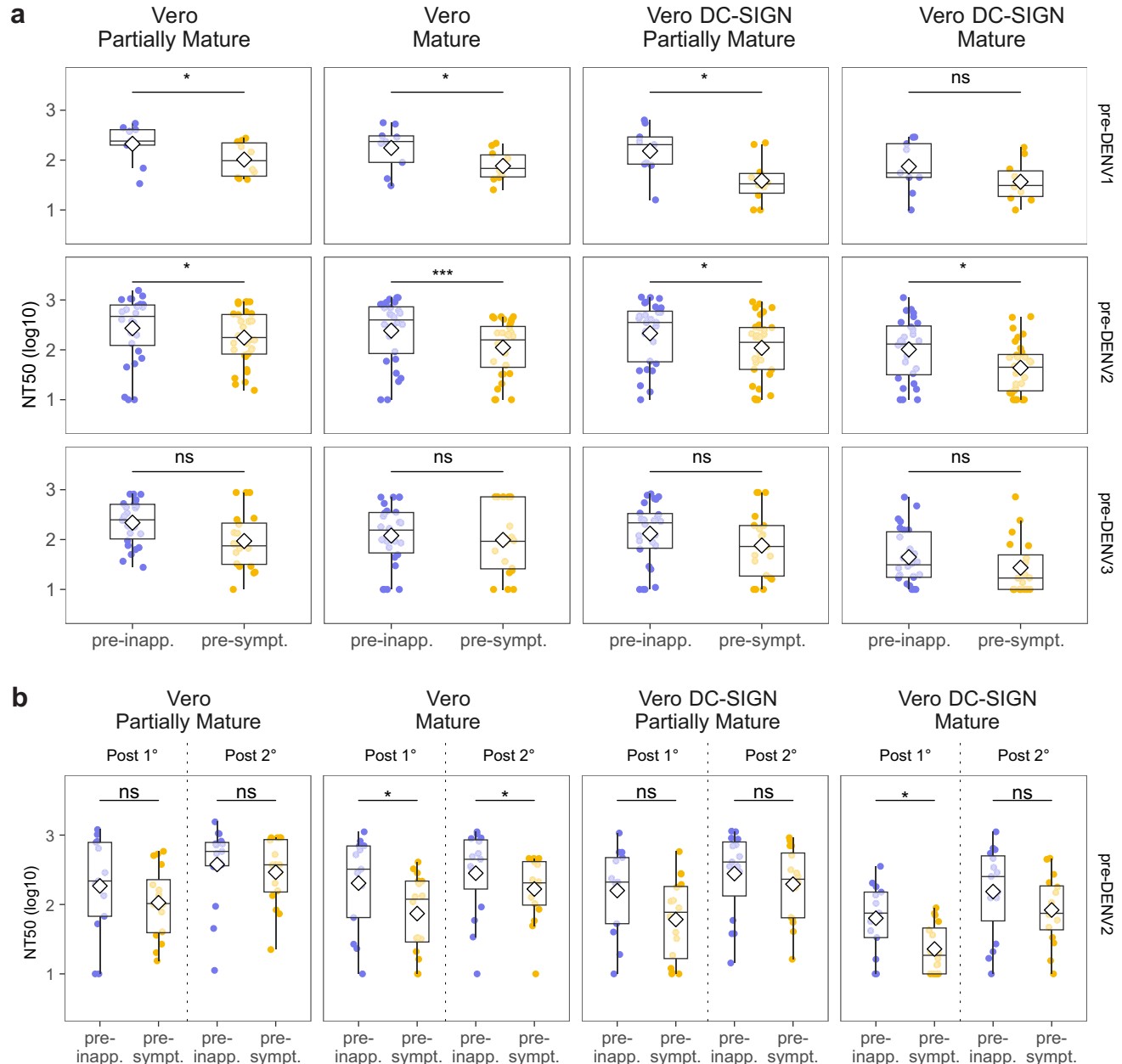

**Fig. 5 | Cross-reactive nAb titers of pre-inapparent and pre-symptomatic DENV infection participants stratified by incoming serotype and infection history.** **a** $NT_{50}$ of pre-inapparent and pre-symptomatic DENV infection groups across FRNT assay conditions stratified by incoming serotype (total pre-DENV1 $n=19$, pre-DENV2 $n=60$, pre-DENV3 $n=46$). **b** $NT_{50}$ of pre-DENV2 infection participants stratified by infection history (Post-1° $n=29$, Post-2° $n=31$). Shown are median $NT_{50}$ (middle line), 25th to 75th percentile (box), and 5th to 95th percentile (whiskers) as

well as the raw data (points). Diamond indicates mean $NT_{50}$. Asterisks indicate Benjamini−Hochberg adjusted $p$-values for Wilcoxon test (two-sided). $p$-values: ns >0.05; *<0.05; **<0.01; ***<0.001; ****<0.0001. Post-1° post-primary infection, Post-2° post-secondary infection, pre-inapp. pre-inapparent DENV infection group (purple), pre-sympt. pre-symptomatic DENV infection group (yellow). Source Data are provided as a Source Data file.

attachment factor for DENV, and expressing it on the surface of Vero cells significantly increases DENV infectivity. DC-SIGN is expressed on dendritic cells (DCs) and monocytes[18–20] and is associated with DENV pathogenesis. Nevertheless, while nAbs that impede DC-SIGN-mediated viral entry could limit viral dissemination and promote the establishment of a rapid and effective cell-mediated immune response[20], this parameter is rarely considered when studying antibody-mediated protection against dengue disease. In this study, we assessed the effect of DC-SIGN expression on Vero cells to measure nAbs able to block the N-glycan-mediated entry pathway. As expected, nAb titers were strongly affected (decreased) by the expression of DC-SIGN, and the reduction in cross-reactive nAb potency was even

greater when assays were performed using mature virions. Similar results were previously published using another flavivirus closely related to DENV, West Nile virus, together with DC-SIGNR expressing-cells[11]. We found that when mature virions are used with Vero DC-SIGN cells, 90% of the subsequent symptomatic infections can be identified with a lower $NT_{50}$ cutoff than with other assay conditions, particularly in post-primary infections. This shows that more serum antibodies are required to neutralize in vitro assays with more physiologically relevant viruses and cells. Thus, anti-DENV nAb titers are likely overestimated when measured in standard Vero cell-based assays and virions with a maturation state different from the virions that antibodies encounter in the bloodstream during infection. These

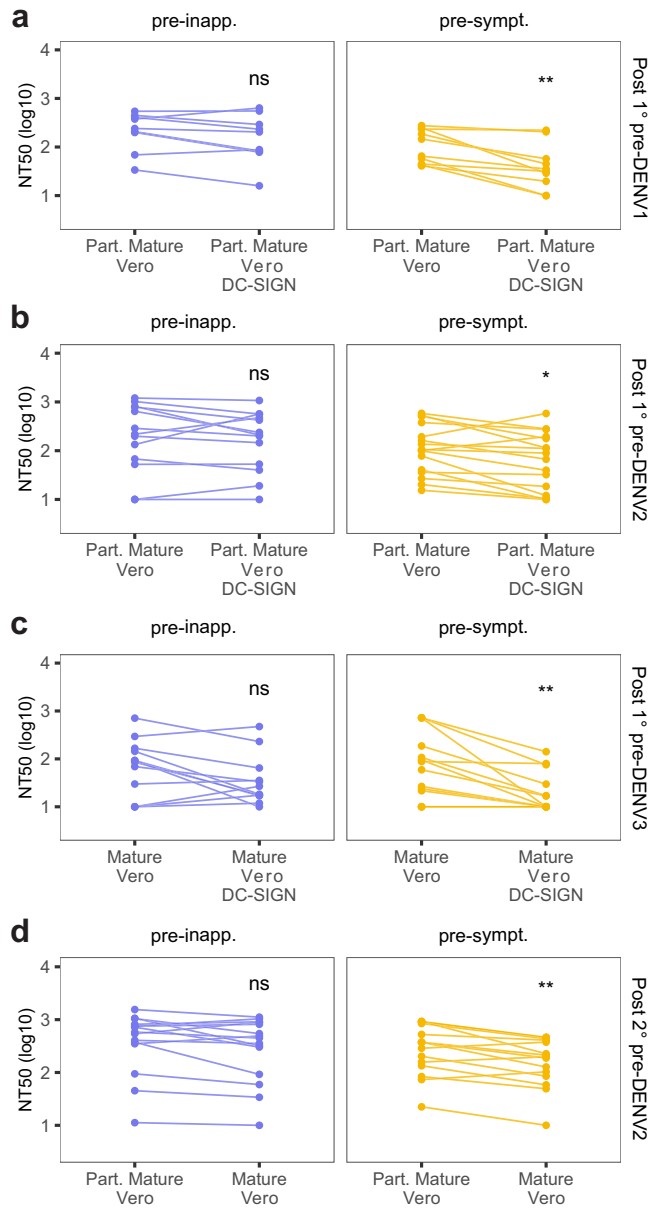

**Fig. 6 | Serotype-specific characteristics of the nAb response associated with DENV infection outcome.** Impact of change in assay condition on the nAb response of pre-inapparent and pre-symptomatic infection participants from the **a** post-primary pre-DENV1 infection group ($n = 19$), **b** post-primary pre-DENV2 infection group ($n = 29$), **c** post-primary pre-DENV3 infection group ($n = 26$), and **d** post-secondary pre-DENV2 infection group ($n = 31$). Asterisks indicate Benjamini–Hochberg adjusted $p$-values for paired Wilcoxon tests (two-sided). $p$-values: ns >0.05; *<0.05; **<0.01; ***<0.001; ****<0.0001. Post-1° post-primary infection, Post-2° post-secondary infection, pre-inapp. pre-inapparent DENV infection group (purple), pre-sympt. pre-symptomatic DENV infection group (yellow), Part. Mature partially mature. Source Data are provided as a Source Data file.

findings highlight the importance of selecting an appropriate cell substrate and virion maturation state to support vaccine development and evaluate efficacy.

Consistent with previous reports[7,34], we observed significantly higher nAb titers in participants before an inapparent DENV infection as compared to before a symptomatic DENV infection when analyzing the entire sample set. However, after stratification by post-primary versus post-secondary DENV infection history, we found that the magnitude of the nAb response only significantly correlated with protection after prior primary infection, although the direction of the

effect was similar for post-secondary infections. Further, we observed differences in association with infection outcome when stratifying by incoming serotype. Higher magnitude of pre-infection nAb titer was associated with protection from subsequent symptomatic infection with DENV1 or DENV2, but not with DENV3 in any of the assay conditions or infection histories. Interestingly, in a recent study investigating potential additional antibody characteristics that could play a protective role, we found that antibody-dependent complement deposition (ADCD) and complement- and antibody-mediated virolysis were associated with protection from symptomatic disease in samples collected prior to a subsequent DENV3 infection[35]. Here, we also found that, while their nAb titer magnitude was similar, the neutralizing activity of post-secondary pre-symptomatic infection individuals was primarily driven by both DC-SIGN-sensitive and maturation state-dependent Abs, while the neutralizing potency of post-primary pre-inapparent infection was not. This suggests that variations in the composition of the antibody repertoire exist in the protected versus unprotected groups and that a difference in nAb quality might be associated with disease outcome.

In addition, while 90% of the subsequent symptomatic infections in our study had pre-infection $NT_{50}$ titers below the $NT_{50}$ threshold identified via ROC analysis, a substantial proportion of pre-inapparent infection individuals did not reach this cutoff either, yet they did not develop a symptomatic infection. This suggests that although nAbs play a role in protection against symptomatic disease, the magnitude of nAb titers is not a robust parameter to differentiate protected individuals from those at risk of developing a symptomatic infection and that other components of the immune response may play an even more important role. This suggests that the measurement of nAb quality and other antibody Fc functions such as ADCD or virolysis[35] may serve as correlates of protection, which could be used in addition to neutralization assays toward the design and evaluation of dengue vaccines, as well as other components of the immune response such as T cells.

The strengths of our study include the unique sample set that allowed us to evaluate the effects of neutralization assay condition on the outcome of DENV infection in natural populations. Samples from diverse epidemics allowed us to observe the distinctions in contribution of cross-reactive nAbs to protection from subsequent symptomatic infection depending on infection history and incoming serotype, which have not been reported previously. We also used clinical isolates representing circulating strains in the study population, which is important because highly passaged reference strains are cell line-adapted and have accumulated mutations that can change their antigenic landscape[10,36]. In contrast, many vaccine studies and research laboratories frequently use WHO reference strains that do not match the genotypic and phenotypic features of the contemporary DENV strains[37,38].

Nevertheless, our study has limitations. First, since we studied sequential DENV infections in natural populations, we evaluated the role of cross-reactive nAbs in protection, but were not able to assess type-specific nAbs, which are often the focus of studies of vaccine efficacy[5]. Nonetheless, many of our findings and overall conclusions are relevant for studies of immune correlates of dengue. Second, as our study population is from Nicaragua over a 15-year period, our analysis is naturally limited to the circulating serotypes and epidemiology of DENV in the timeframe and area of study. Further, we were not able to address the impact of different sequences of infections in the composition of the nAb response due to small sample sizes of specific serotype sequences. Third, although anti-DENV antibodies can also have infection-enhancing properties, we do not measure this here since we focus on studying the neutralizing capacity of antibodies using cell substrates that do not express Fcγ receptors that can mediate antibody-dependent enhancement[39]. Lastly, our current sample size is under-powered for the predictive model of risk of symptomatic

disease for incoming DENV1 or DENV3 infection. Nonetheless, the observed differences in effect sizes imply that for achieving a power of at least 80% we would need only 28 pre-DENV1 infection individuals, in contrast with 231 pre-DENV3 infection individuals and a total of 222 individuals in the post-secondary infection group. These differences in the required sample size are evidence of the potential negligible role of nAbs in protection against DENV3 symptomatic infection and upon a post-secondary infection.

In conclusion, we have identified serotype-specific differences in the association of nAbs with protection against symptomatic infection. Higher $NT_{50}$ titers were associated with protection against symptomatic DENV infection in participants with one prior DENV infection, but only against DENV2 in those with multiple prior infections. nAbs were correlated with protection from symptomatic DENV1 and DENV2 infections but not DENV3. Thus, the unique features of our cohort sample set allowed us to identify the role of infection history in shaping the protective potency of nAbs, as well as the relevance of the incoming serotype, ultimately revealing the heterogeneity and the complexity of immune interactions among DENV serotypes. Comparing different assay conditions, we found that maturation state- and DC-SIGN-sensitive nAbs may be less protective against symptomatic DENV infection. More studies are needed to assess the effects of these variables, as well as the impact of WHO reference strains compared to clinical isolates, in the context of vaccine studies. Finally, although nAbs correlate with protection under specific circumstances, our findings reveal that they do not capture the full breadth of the protective immune response, thus implying an important role for other aspects of the antibody and broader immune response in generation and evaluation of protection from symptomatic dengue. Incorporating a broader range of immune parameters in serological assays will enable a more comprehensive evaluation of immune correlates of protection.

## Methods

### Ethics statement
The human participants protocol for the Pediatric Dengue Cohort Study (PDCS) was reviewed and approved by the Institutional Review Boards (IRB) of the University of California, Berkeley (2010-09-2245), the University of Michigan (HUM00091606), and the Nicaraguan Ministry of Health (CIRE-09/03/07-008). Parents or legal guardians of all participants provided written informed consent, and participants 6 years of age and older provided assent. All participants live in District II, Managua, Nicaragua, the catchment area for the Health Center Socrates Flores Vivas (HCSFV).

### Samples and study population
The PDCS is an ongoing open prospective cohort of ~4,000 children 2–17 years old that was initiated in 2004 in Managua, Nicaragua[6,25]. Blood samples are collected annually, and additional samples are collected from symptomatic suspected dengue cases at the acute (0–6 days post-onset of symptoms) and convalescent (14–21 days post-onset of symptoms) phase. DENV infections are evaluated and recorded yearly by comparing healthy annual serum/plasma samples collected in two consecutive years side-by-side by inhibition enzyme-linked immunosorbent assay (iELISA)[2]. A ≥ 4-fold increase in iELISA titer between annual samples is considered as indication of an infection[40].

Symptomatic dengue cases were confirmed by detection of DENV RNA by RT-PCR, real-time RT-PCR and/or virus isolation in the acute-phase sample; seroconversion of DENV-specific immunoglobulin M antibodies in paired acute- and convalescent-phase samples; or a ≥ 4-fold increase in antibody titer by iELISA between acute and convalescent sera[2,41–43]. The serotype of symptomatic infections was determined by RT-PCR or real-time RT-PCR[41,44,45]. Inapparent infections were identified with a ≥ 4-fold increase in iELISA titer among participants who did not present with illness to the study Health Center

between annual samplings. In a subset, the serotype responsible for the inapparent infections that occurred before 2016 was identified by neutralization assays (DENV1-4) and/or a multiplex Luminex-based assay (E domain III [EDIII] of DENV1-4 and ZIKV) by comparing the antibody profile in the annual samples collected before and after the infection. Inapparent infections detected in 2016-2019 were attributed to DENV2 based on laboratory and epidemiological data showing the circulation of only DENV2 in Nicaragua during these years.

We selected 125 DENV-immune children who subsequently developed an inapparent infection ($n = 64$) or symptomatic infection ($n = 63$) and analyzed their pre-infection plasma samples. Samples obtained after a single DENV infection are referred to as post-primary infection samples, while samples obtained after two or more DENV infections are referred to as post-secondary infection samples (Fig. 1).

The criteria for sample selection were 1) for pre-symptomatic infection: an annual sample from a child post-primary or post-secondary DENV infection collected prior to a subsequent hetero-typic symptomatic DENV infection that was serotyped by RT-PCR, 2) for pre-inapparent infection: an annual sample from a child post-primary or post-secondary DENV infection collected prior to a subsequent heterotypic inapparent DENV infection and an annual sample collected right after the given infection to confirm the infecting serotype. Group assignment was performed according to infection outcome (symptomatic or inapparent infection), prior infection history (post-primary or post-secondary DENV infection), and infecting serotype.

Samples in question are from pediatric participants in our long-term cohort study and are therefore very limited in volume. Thus, these materials are subject to restriction based on limited availability. Any transfer of materials needs to be approved by the IRB and by Dr. Harris and Nicaraguan researchers. For more information, please contact Dr. Eva Harris (eharris@berkeley.edu) and/or the committee for the Protection of Human Subjects at University of California Berkeley (510-642-7461; ophs@berkeley.edu).

### Cell lines
Vero cells (ATCC CCL-81), Vero cells stably expressing human furin (Vero-Furin; kind gift of Victor Tse and Ralph Baric, University of North Carolina (UNC), Chapel Hill)[46], and Vero cells expressing DC-SIGN (Vero DC-SIGN) were maintained in minimum essential medium (MEM; Gibco) with 5% fetal bovine serum (FBS) supplemented with 1X non-essential amino acids (Gibco), 1X sodium pyruvate (Gibco), and 1X penicillin-streptomycin (Gibco) at 37 °C in 5% $CO_2$.

### Generation of Vero DC-SIGN cell line
Vero cells were transduced using lentivirus to express DC-SIGN (CD209). DC-SIGN surface expression was confirmed by flow cytometry using a PE-conjugated mouse anti-CD209 antibody (1:1000, BD Bioscience, BDB561765) for cell surface staining. A stock of Vero cells expressing surface DC-SIGN receptors at >95% was generated by cell sorting using the BD InfluxTM cell sorter and the BD FACSDivaTM 8.0.1 software (BD Biosciences).

### Virus production
DENV1 (5575.10a1SPD1), DENV2 (8891.12a1SPD2) and DENV3 (6629.10a1SPD3) are clinical isolates from Nicaragua, isolated on C6/36 cells and passaged twice in Vero cells prior to production of working stocks. Partially mature and mature viruses were grown in Vero and Vero-Furin cells, respectively, in the same growth medium described above. Virions were harvested 5 days post-infection and frozen at 80 °C until use.

### Viral titer
Infectious titers were determined using a standard focus-forming assay[37]. Briefly, Vero and Vero DC-SIGN cells were infected with serial

dilutions of virus and incubated at 37 °C for 42–48 hours (h) depending on the virus strain. Cells were fixed, permeabilized with 0.5% Triton for 6 min, and stained with primary mouse anti-FL monoclonal antibody (mAb) 4G2 for 1.5 h (1:1000, Biomatik. AB00230-2.0, Clone D1-4G2-15), washed 2 times with PBS BSA 1%, and incubated with a goat anti-mouse secondary antibody conjugated to horseradish peroxidase for 1 h at room temperature (1:3000, Biolegend, 405306). Foci were developed using True Blue HRP substrate (KPL) and were counted using an CTL Immunospot analyzer (Cellular Technology, Ltd.). The titer in focus forming units (FFU) per ml was then calculated based on the average of replicates.

### Viral genome quantification and calculation of infectivity

Viral RNA (vRNA) was extracted from virus particles using the QIAmp kit (QIAGEN). The PCR standard curve used for the quantification of DENV copy genome was obtained with an amplicon containing a synthetic cDNA encompassing a fragment of the 3′UTR region (nucleotides 10361–10660) from DENV2 NI15. Pan-DENV primers (F – TTG AGTAAACYRTGCTGCCTGTAGCTC; R - GAGACAGCAGGATCTCTGG TCTYTC,) were used to amplify the 3′UTR region encompassing nucleotides 10361–10660 of the genome of DENV clinical isolates used in this study. To assess the proportion of infectious viral particles in each of the virus stocks, viral titers obtained in different cell lines were plotted against the number of RNA copies present in each dilution of virus studied. Specific infectivity was calculated as the number of RNA molecules required for each infection event (vRNA:FFU).

### Assessment of virion maturation state

For each serotype, equal numbers of vRNA genome equivalents (GE) of partially mature and mature viral stocks were diluted in 6X Laemmli sample buffer, boiled at 95 °C for 5 min, and subjected to SDS-PAGE electrophoresis on a 4–20% gradient gel. Proteins were then transferred to a PVDF membrane and blocked using a blocking buffer containing 5% non-fat milk in PBS + 0.1% Tween-20 (PBS-T). The membrane was incubated with a purified human anti-E FL mAb (1M7) and anti-prM mAbs (2G3, 1E23 and 2H21) at 0.1 ug/ml in blocking buffer at 4 °C overnight. Antigen-antibody complexes were detected using donkey anti-human IgG conjugated to horseradish peroxidase (HRP; 1:5000, Biolegend) in blocking buffer for 1 h at room temperature. After washing, membranes were developed in Supersignal West Pico PLUS Chemiluminescent Substrate (ThermoFisher). Western blot images were captured with a ChemiDoc XRS+ system (Bio-Rad). The pixel intensity of individual bands was measured using FIJI software, and relative maturation was calculated by using the following equation: $(prM_{Vero\ Furin}/E_{Vero\ Furin})/(prM_{Vero}/E_{Vero})$.

### Neutralization assay

The focus reduction neutralization test (FRNT) assay was conducted using Vero or Vero DC-SIGN cells. Briefly, cells were seeded in a 96-well plate one day before infection. Serum/plasma or mAbs were serially diluted and mixed with an equal inoculum of DENV (equivalent to 40–50 FFU/well for Vero and 150–200 FFU/well for Vero DC-SIGN cells) at a volume ratio of 1:1 to form immune complexes. The virus-antibody mixture was incubated for 1 h at 37 °C in 5% $CO_2$. Cell substrate growth medium was removed, and the virus-antibody mixture was added to Vero or Vero DC-SIGN cells and incubated for 1 h at 37 °C in 5% $CO_2$. Subsequently, a 0.6% carboxymethylcellulose overlay was added, and the plates were incubated for 48 h. Foci were developed and counted as described above. Foci counts were fitted using a variable slope dose response curve using Graphpad, and neutralizing antibody titer ($NT_{50}$) values were calculated with constrained top and bottom values of 100 and 0, respectively. Percent relative infection was calculated as a ratio of foci counts in each serial dilution to the foci counts from the final serial dilution of each sample. All samples and mAbs were run in duplicate with reported values required to have an

$R^2 > 0.85$ and a Hill slope > 0.5. Neutralization titers were measured against the heterotypic incoming serotype of each participant.

### Statistical analysis

All statistical analyses were conducted using R (R Foundation for Statistical Computing, version 4.5.0) and the following packages: readr v2.1.4, dplyr v1.1.1, ggplot2 v3.4.2, rstatix v0.7.2, ggpubr v0.6.0, tableone v0.13.2, pROC v1.18.0, stats v4.2.2, oddsratio v2.0.1, pwr v1.3-0, pscl v1.5.5.

Participants' characteristics were compared using ANOVA for numerical or chi-square for categorical variables (two-sided). No adjustments nor multiple comparisons were performed.

$NT_{50}$ values were obtained using GraphPad software as indicated above. Normal distribution of data was tested using the Shapiro-Wilk test. Adjusted p-values and statistical significance of nAb titer magnitude and differential $NT_{50}$ comparisons were computed using the Wilcoxon test with Benjamini–Hochberg correction. Adjusted *p*-values and statistical significance of nAb potency across neutralization assay conditions were computed using the paired Wilcoxon test (two-sided) with Benjamini–Hochberg correction to identify changes in antibody quality.

Receiver Operator Curve (ROC) analysis was performed using the pROC package (v.1.18.0). The pooled area under the curve (AUC), sensitivity and specificity to detect a subsequent symptomatic infection was estimated across different assay conditions. The cut-off of the assay was defined as the $NT_{50}$ value under which at least 90% of the subsequent symptomatic infections were detected (sensitivity). The specificity corresponded to the proportion of inapparent infections above this same value. A stratified analysis was also conducted to evaluate the differential performance of the assay conditions according to incoming infecting serotype (DENV1, DENV2 and DENV3). Finally, a logistic regression model was constructed using the stats package v.4.2.2 and the oddsratio package v. 2.0.1 in R studio. We assumed a binomial distribution of the response variable and a significance level (alpha) of 5%. We estimated the odds ratio (OR) of developing a subsequent symptomatic versus inapparent infection given an increment of $10^2$ in the $NT_{50}$. Power calculations were conducted using the pwr package v. 1.3-0. The package pscl v 1.5.5 was used for estimating the McFadden's pseudo R-squared of each model.

Graphs and schemes were made using ggplot2 package and BioRender.

### Reporting summary

Further information on research design is available in the Nature Portfolio Reporting Summary linked to this article.

## Data availability

Datasets generated and/or analyzed during the current study are included within the main manuscript or are appended as supplementary data. Data analysis was performed using R and packages that are publicly available, no custom code was generated. Source data are provided with this paper.

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

## Acknowledgements

We are grateful for the PDCS participants and their families. We appreciate the hard work and dedication of the past and present members of the study team at the Centro de Salud Sócrates Flores Vivas, the Sustainable Sciences Institute in Nicaragua, and the Laboratorio Nacional de Virología in the Centro Nacional de Diagnóstico y Referencia. We thank Scott Biering and Richard Ruan for generating the Vero-DC-SIGN cells.

## Author contributions

S.B. and E.H. conceived the project. S.B. and A.G. designed the experiments. A.G., E.D., R.M.-H. and T.S. carried out the experiments. J.V.Z., J.A.R. managed the clinical and laboratory databases and organized the data for analysis. G.K. and A.B. directed the cohort study. S.B., A.G., E.D., J.C.O., L.K., A.S. and E.H. contributed to the analysis and interpretation of the results. E.H. obtained funding. S.B., A.G., J.C.O. and E.H., wrote the manuscript with input from all authors. All authors provided critical feedback and helped shape the research, analysis and manuscript. This work was supported by the National Institute of Allergy and Infectious Diseases/National Institutes of Health (NIAID/NIH) via grant P01AI106695 (E.H.). The PDCS study was supported by NIAID/NIH grants U01AI153416 (E.H.) and U19AI118610 (E.H.), R01AI099631 (A.B.), the Pediatric Dengue Vaccine Initiative grant VE-1 (E.H.) from the Bill and Melinda Gates Foundation, and NIH subcontract HHSN2722001000026C (E.H.). L.C.K. was supported by the Intramural Research Program of NIAID/NIH.

## Competing interests

The authors declare no competing interests.

## Ethics approval

This research is embedded in a long-term 35-year close collaboration with Nicaraguan colleagues that has included multiple aspects of scientific capacity building and enhancement, as well as supporting improvement of physical research infrastructure. With regards to this study, local Nicaraguan researchers were included throughout the research process – in study design, study implementation, data ownership, and authorship of publications. Four of the authors on this publication are from the local team in Nicaragua. The research is locally relevant, and this was determined in collaboration with local partners in Nicaragua. We discussed roles and responsibilities amongst collaborators ahead of the research, and we conducted capacity-building in the form of additional on-site training of local researchers in virus production and neutralization assays. The parent cohort study has been approved initially and every year thereafter (currently in its 20th year) by a local ethics review committee. This research does not result in stigmatization, incrimination, discrimination, or otherwise personal risk to participants nor any health, safety, security or other risk to researchers. Benefit sharing measures related to transfer of biological materials been discussed with our partners in Nicaragua. Finally, we have taken local and regional research relevant to our study into account in citations.
