## [Peer Review File · Nature Communications]

REVIEWER COMMENTS

Reviewer #2 (Remarks to the Author):

This is an important and well done study and paper. However I found the results hard to follow at times, and so I have a few comments on helping the reader understand the results below, but would suggest a general look at the results to see where clarity can be increased.

Introduction:

“there is an increased risk..” suggest rephrasing as the link between the first and second parts of the sentence is not entirely clear.

The introduction is clear, but quite long. Would suggest- but it isn't necessary-to reduce a little.

The last section of the introduction summarises the results I would suggest removing this.

Results:

Is the “secondary” and “>=2 infections” referring to the same thing, if so I would use the same wording. If not- suggest making it clear what the difference is here, and why different measurements are used. “post-primary” and “post-primary” is then used in the results- is this a subset of the above- if so be good to mention how many in each of these groups in the methods too (and being clear what these mean).

I think it would also be helpful to have this broken down by serotype too- it is hard to follow in the results section on serotypes which groups had data and which didn't.

“Thus, our findings reinforce that increasing virion maturation state and the presence of

DC-SIGN significantly decrease the estimation of serum DENV-neutralizing potency.” The relationship shown is not clear from this sentence- suggest slight rewording.

NT50 “established threshold” is this one you've defined, as in the following sentence referred to as cut-off value? or something previously defined. Suggest rewording/using the same wording if it is the same thing. This is necessary for understanding the following results- so I'll comment on those next time round.

“Given the impact of virion maturation state and cell substrate on NT50 performance as a predictor of subsequent symptomatic infection” be clear which of your results you are referring to here..

“we did not observe any

significant difference in the magnitude of nAb titers from post-primary pre-inapparent infection

individuals (i.e., protected) and post-secondary pre-symptomatic infection individuals (i.e., not protected) across all four conditions tested (Fig. 5A,B).” I can’t follow why this comparison was done, please add more information.

“Pre-infection neutralizing antibody composition differentially impacts outcome by DENV serotype and infection history” how is the section different to some of the previous sections.

As evidenced by some of my points, I found the different sections of the results hard to follow and many of the results hard to understand. I would suggest rewording to be clear what is being compared to what and why and if similar comparisons are made, but with different methods I’d suggest putting these together in the results.

Reviewer #3 (Remarks to the Author):

General comments:

In this study the authors have investigated the cross reactive neutralizing antibody responses to different Dengue serotypes in a pediatric cohort to establish antibody correlates of protection, which has implications in vaccine development. In particular, they assessed Nab response prior to subsequent asymptomatic/symptomatic infection by Denv1/2/3 using plasma samples, comparing neutralization potency of partially mature and fully mature particles on Vero cells and Vero cells expressing DC-SIGN. The results indicate that the level of protection is dependent on prior immune status, the serotype of the subsequent infection and assay conditions. While these data provide insights into the variability of antibody responses elicited (reinforcing information from existing literature) and parameters that should be considered during vaccine development, the study itself does not provide any new information or mechanistic insights into immune responses to Dengue infection. It is also fairly limited in scope and is entirely correlative in the current version.

Specific comments:

- Some of the strengths of this study are as follows: the use of the pediatric cohort is clearly a major strength of this study. They also confirmed previous findings on the importance of prior immunity toward developing asymptomatic infection as well as on the importance of the serotype in disease outcomes, which is an important point. Data showing breadth of protection, both for partially mature and fully mature particles is also important.

There are several gaps in this study as outlined below, which makes it difficult to draw any conclusions from the data presented:

- the study was conducted in DC-SIGN expressing Vero cells and therefore does not take into account the FcγR mediated entry and ADE, which is expressed highly in DCs and monocytes that are the physiological targets of Dengue. It might have been better to use DCs to do a more systematic analysis of DC-SIGN, FcγR and other potential receptors by using individual knock-outs.

- The threshold for specificity and sensitivity are not well described, whilst being important measures for assessing their data.

- In general the magnitude of the effects are very low. Given that the sample size is fairly low, it is slightly difficult to gauge whether these differences are biologically significant. The choice of statistics are also not clear – it would be useful to explain why a paired Wilcoxon test was used in Fig 1.

- Similarly, Fig 6C shows a more pronounced difference in pre-inapp sample compared to pre-symptomatic ones, yet the difference is non-significant.

- Some figure legends need to be amended – e.g. Figure 5C is not described.

- Similarly, Fig 3B description needs to be clarified.

- It would also be important to include assessment of cross-reactive antibodies between different serotypes (i.e post 1st DENV1 antibodies vs pre-inapparent DENV3 tested with DENV3- since the samples are already available).

NCOMMS2327509: Response to Reviewers

EDITOR :

- *Thank you again for submitting your manuscript "The association of neutralizing antibodies with protection against symptomatic dengue virus infection varies by serotype, prior immunity, and assay condition" to Nature Communications. We have now received reports from 2 reviewers and, after careful consideration, we have decided to invite a major revision of the manuscript*

R/ Thank you. We have addressed all the comments raised by the reviewers below.

- *As you will see from the reports copied below, the reviewers raise important concerns. We find that these concerns limit the strength of the study, and therefore we ask you to address them with additional work. Without substantial revisions, we will be unlikely to send the paper back to review. In particular, a revised manuscript will need to provide additional data and/or analysis to address concerns about insufficient analysis of cross-reactive antibodies and used statistics. We will also need the concern raised about the model used in the study to be addressed.*

R/ The Results section of the manuscript has been extensively edited to address all the comments made by the reviewers. We included additional analyses and a revised version of Figure 4 to improve clarity regarding our results. We also included a description of the rationale for the statistics implemented in the Methods section, as well as an additional supplementary table and figure that contain relevant information about the effect size and power for each model plus data supporting the test selected for hypothesis testing.

- *Please note that we generally require a final paragraph in the Introduction section that summarizes the main findings, so there is no need to remove this to address reviewer #1's comment on this. Please ensure that these and all other concerns raised by our reviewers are addressed in full in a revised manuscript.*

R/ The paragraph in question was retained.

Point by point response:

REVIEWER #2:

- *This is an important and well done study and paper. However, I found the results hard to follow at times, and so I have a few comments on helping the reader understand the results below, but would suggest a general look at the results to see where clarity can be increased.*

R/ Thank you. We have extensively edited the results section of the manuscript to improve clarity and make the sections and flow easier to follow.

- Introduction:

- › *“there is an increased risk..” suggest rephrasing as the link between the first and second parts of the sentence is not entirely clear. The introduction is clear, but quite long. Would suggest- but it isn’t necessary-to reduce a little.*

R/ We have removed the sentence in question because it was not clear and to make the introduction shorter. The flow of the introduction has been maintained, as well as the last paragraph per the editor’s request.

- Results:

- › *Is the “secondary” and “>=2 infections” referring to the same thing, if so I would use the same wording. If not- suggest making it clear what the difference is here, and why different measurements are used. “post-primary” and “post-primary” is then used in the results- is this a subset of the above- if so be good to mention how many in each of these groups in the methods too (and being clear what these mean).*

R/ In our manuscript, primary DENV infections are defined as having a single previous DENV infection, while secondary DENV infection is defined as having had two or more previous DENV infections.

To address the Reviewer’s comment, we have consolidated the use of “primary” and “secondary” DENV infection through the manuscript. We have clarified this in the results section:

Lines 336 to 338: “At the time of sample collection, 75/127 participants had experienced a single DENV infection (primary infection), and 53/127 participants had experienced two or more DENV infections (secondary infection)”

AND

Lines 342 to 344: “Samples collected from participants with prior primary infection will be referred to as “post-primary” (Post-1°), while samples collected from participants who previously had had two or more DENV infections will be referred to as “post-secondary” (Post-2°).”

We also included a better description of this definition and the sample size of each group in the Methods (below), and we included a Figure (Fig. 1) describing the study design:

Lines 152 to 156: “We selected 127 DENV-immune children who subsequently developed an inapparent infection (n=64) or symptomatic infection (n=63) and analyzed their pre-infection plasma samples. Samples obtained after a single DENV infection are referred to as post-primary infection samples, while samples obtained after two or more DENV infections are referred to as post-secondary infection samples (**Figure 1**)”

- › *I think it would also be helpful to have this broken down by serotype too- it is hard to follow in the results section on serotypes which groups had data and which didn’t.*

R/ We agree. We have included this information as part of Table 3

- › *“Thus, our findings reinforce that increasing virion maturation state and the presence of DC-SIGN significantly decrease the estimation of serum DENV-neutralizing potency.” The relationship shown is not clear from this sentence- suggest slight rewording.*

R/ We agree. To provide clarity regarding the interpretation of our results, we have edited this section of the results. In particular, we re-worded the sentence in question as follows:

Lines 299 to 307: “Our results demonstrate that overall, NT₅₀ values were significantly lower in assays performed with mature virions compared to partially mature virions, indicating that serum antibodies had a lower potency for neutralizing the incoming DENV virion in its mature form. Additionally, NT₅₀ values were lower in assays performed on Vero DC-SIGN cells compared to Vero cells (Fig. 2), indicating a decrease in the potency of nAbs under conditions that facilitate glycan-mediated DENV entry. Notably, the most significant difference in median NT₅₀ values was observed between assays performed using Vero cells infected with partially mature virions (reference assay condition, NT₅₀=202) and those using Vero DC-SIGN cells infected with mature virions (NT₅₀=42)”

- › *NT50 “established threshold” is this one you’ve defined, as in the following sentence referred to as cut-off value? or something previously defined. Suggest rewording/using the same wording if it is the same thing.*

R/ The cut-off value was established using a ROC analysis to analyze the performance of each assay condition for discriminating between subsequent symptomatic and inapparent infections. We clarified the definition of the cut-off value (Lines 234 to 239, Methods section, and Lines 453 to 457, Results section). And we also changed the word “threshold” to “cut-off value” in order to keep it consistent across the manuscript. We added:

Lines 234 to 239: “Receiver Operator Curve (ROC) analysis was performed using the pROC package (v.1.18.0) in R studio (v. 2022.12.0+35). The pooled area under the curve (AUC), sensitivity, and specificity to detect a subsequent symptomatic infection was estimated across different assay conditions. The cut-off of the assay was defined as the NT₅₀ value under which at least 90% of the subsequent symptomatic infections were detected (sensitivity). The specificity corresponded to the proportion of inapparent infections above this same value.”

Lines 453 to 457: “We performed a ROC analysis to compare the assay conditions in terms of their sensitivity and specificity (Table 3). The cut-off of the assay was defined as the NT₅₀ value under which at least 90% of the subsequent symptomatic infections were detected (sensitivity). The specificity corresponded to the proportion of inapparent infections above the cut-off value.”

To improve the interpretability of our results, we also included the following:

Lines 462 to 468: “For the assay performed on partially mature virions on standard Vero cells, which is the assay used in most clinical trials for vaccine testing, the NT₅₀ cut-off for achieving a sensitivity of at least 90% was 898. Nevertheless, at this titer, the specificity was only 8%. This means that, using this assay, although more than 90% of the individuals had an NT₅₀ below 898 before developing a symptomatic DENV infection, 92% of the

individuals that subsequently developed an inapparent infection were also below this cut-off and yet were protected from disease.”

- › “Given the impact of virion maturation state and cell substrate on NT₅₀ performance as a predictor of subsequent symptomatic infection” be clear which of your results you are referring to here.

R/We thank the Reviewer for pointing out this section that needed clarification and have added a concluding paragraph in the preceding section summarizing the results to which we are referring:

Lines 364 to 370: “Collectively, these findings highlight that the association between nAb titer magnitude and protection against symptomatic DENV infection depends on the infection history of participants. Moreover, they suggest that while robust production of cross-reactive nAbs (magnitude) is a critical factor in protection against a second DENV infection, it appears insufficient to confer protection against subsequent infections. This implies that other nuanced aspects within or beyond the nAb response might contribute more to protection in individuals with secondary DENV infection histories.”

We also edited the sentence to provide more clarity.

Lines 371 to 379: “In light of these observations, we leveraged the different assay conditions to gain insight into the composition of the nAb response. To do so, we compared the neutralizing potency of nAbs from each participant across paired assay conditions. Specifically, comparison of the NT₅₀ measured using partially mature versus mature virions was used to evaluate the contribution of antibodies that are maturation state-dependent, while comparison of NT₅₀ measured on standard Vero versus Vero-DC-SIGN cells was used as a proxy to evaluate whether neutralization was mediated by antibodies capable of neutralizing DENV through N-glycan-mediated entry.”

- › “we did not observe any significant difference in the magnitude of nAb titers from post-primary pre-inapparent infection individuals (i.e., protected) and post-secondary pre-symptomatic infection individuals (i.e., not protected) across all four conditions tested (Fig. 5A,B).” I can’t follow why this comparison was done, please add more information.

R/ Participants with a prior primary DENV infection had a significantly higher NT₅₀ value in the pre-inapparent group compared to the pre-symptomatic group across all four conditions (Fig. 3B). However, in the post-secondary group, both pre-inapparent and pre-symptomatic participants exhibited high NT₅₀ titers with no significant difference in titer magnitude.

Notably, we observed that the titers in participants from the post-secondary pre-symptomatic infection group were as high as the nAb titers observed in participants with one prior infection who were *protected* against a subsequent symptomatic infection (Post-primary pre-inapparent group) (Fig. 4B). This shows that high nAb titers alone are not sufficient for conferring protection, implying that other aspects of the nAb response beyond magnitude, or even beyond neutralization potency, may play a role in protection after secondary DENV infection.

Given this observation, we wanted to explore whether the contribution of nAbs that are maturation state-dependent or the contribution of nAbs capable of neutralizing the N-glycan-mediated entry of DENV were different across these groups. To address this question, we compared the NT₅₀ titers of maturation state-sensitive vs. DC-SIGN-sensitive nAbs.

We clarified this point as follows:

Lines 345 to 356: “In participants with a prior primary DENV infection, we observed significantly higher NT₅₀ values in the pre-inapparent group compared to the pre-symptomatic group across all four conditions (Fig. 3B). This suggests that high titers of cross-reactive nAbs are associated with protection from disease upon a second DENV infection. However, in the post-secondary group, both pre-inapparent and pre-symptomatic participants exhibited high NT₅₀ titers with no significant difference in titer magnitude (Fig. 3B). Notably, we observed that the titers in participants from the post-secondary pre-symptomatic infection group were as high as the titers observed in participants with one prior infection who were protected against a subsequent symptomatic infection (Post-primary pre-inapparent group) (Fig. 4B). This showed that high nAb titers alone are not sufficient for conferring protection, implying that other aspects of the nAb response beyond magnitude, or even beyond neutralization potency, may play a role in protection after secondary DENV infection.”

Lines 380 to 392: “As previously mentioned, we did not observe any significant difference in the magnitude of NT₅₀ from post-primary pre-inapparent infection individuals (i.e., protected) and post-secondary pre-symptomatic infection individuals (i.e., not protected) across all four conditions tested. Consequently, we found these two groups particularly intriguing and compared them, along with post-secondary pre-symptomatic individuals, to investigate whether variations in nAb composition were associated with different outcomes (Fig. 4A, B).”

- › *“Pre-infection neutralizing antibody composition differentially impacts outcome by DENV serotype and infection history” how is the section different to some of the previous sections.*

R/ In this section, we addressed the impact of the nAb composition depending on the incoming serotype. In this way, we aimed to determine whether the overall observation of the combined analysis was similar in participants who subsequently experienced an infection by DENV1, DENV2, or DENV3.

REVIEWER #3:

- *General comments:*

In this study the authors have investigated the cross reactive neutralizing antibody responses to different Dengue serotypes in a pediatric cohort to establish antibody correlates of protection, which has implications in vaccine development. In particular, they assessed Nab response prior to subsequent asymptomatic/symptomatic infection by Denv1/2/3 using plasma samples, comparing neutralization potency of partially mature and fully mature particles on Vero cells and Vero cells expressing DC-SIGN. The results indicate that the level of protection is dependent on prior immune status, the serotype of the subsequent infection and assay conditions. While these data provide insights into the variability of antibody responses elicited (reinforcing information from existing literature) and parameters that should be considered during vaccine development, the study itself does not provide any new information or mechanistic insights into immune responses to Dengue infection. It is also fairly limited in scope and is entirely correlative in the current version.

R/ While we appreciate the Reviewer's comments on this matter, *we believe our manuscript contributes new insights to the topic, as detailed below.* Our work assessed the nAb response prior to subsequent inapparent/symptomatic infection by DENV1, DENV2 or DENV3 using pre-infection plasma samples and investigated the association of the nAb composition, in terms of maturation-sensitive and DC-SIGN-sensitive nAbs, with infection outcome in children with different infection histories.

We believe that our work provides several pieces of information not described before:

First, although previous reports have shown that nAbs elicited after a DENV infection protect against a subsequent symptomatic DENV infection, our findings indicate that the magnitude of cross-reactive nAbs plays a protective role in individuals who had only one single prior DENV infection. However, in individuals who have experienced two or more DENV infections, the magnitude of pre-existing nAbs is not enough to confer protection from a subsequent symptomatic infection.

Second, we show that although the magnitude of cross-reactive nAbs after 2 or more infections is comparable between children who subsequently developed inapparent versus symptomatic infections, the *composition* of the nAb repertoire is different between these two groups. Specifically, we found that protection after 2 or more infections is not driven by magnitude but rather by maturation state-sensitive and DC-SIGN-sensitive nAbs.

Third, we show that in the context of natural infection, the effect size of the protection conferred by cross-reactive nAbs depends on the incoming serotype, and that nAbs targeting mature virions or blocking entry to cells expressing DC-SIGN are relevant for protection against an incoming DENV2, but not against an incoming DENV1 or DENV3 infection.

Fourth, though it is well-known that assay conditions influence the outcome of neutralization assays, our work specifies the effect of two well-controlled parameters in particular not only on NT₅₀ values but on their association with infection outcome in human populations.

All together, these findings show that nAb titer magnitude and composition vary depending on DENV infection history and confer protection from symptomatic DENV infection in a serotype-dependent manner, a fact that has been suggested but not proven before, especially in the context of natural infection.

- *Specific comments:*

- *Some of the strengths of this study are as follows: the use of the pediatric cohort is clearly a major strength of this study. They also confirmed previous findings on the importance of prior immunity toward developing asymptomatic infection as well as on the importance of the serotype in disease outcomes, which is an important point. Data showing breadth of protection, both for partially mature and fully mature particles is also important.*

R/ Thank you; we appreciate the Reviewer's words of support.

There are several gaps in this study as outlined below, which makes it difficult to draw any conclusions from the data presented:

- *the study was conducted in DC-SIGN expressing Vero cells and therefore does not take into account the FcγR mediated entry and ADE, which is expressed highly in DCs and monocytes that are the physiological targets of Dengue. It might have been better to use DCs to do a more systematic analysis of DC-SIGN, FcγR and other potential receptors by using individual knock-outs.*

R/ We thank the Reviewer for this comment. While we agree that the questions raised by the Reviewer are important, the goal of our study was to characterize the nAb titers *in relation to infection outcome in human populations* using the same cell substrate as the current standard assays in a way that would allow us to identify the role of DC-SIGN. By using Vero cells expressing this attachment factor or not, we wanted to interrogate the variation in neutralizing potency when a protein that can facilitate glycan-mediated entry is present on the cell substrate.

Additionally, we wanted to characterize the pre-existing nAbs in terms of the protection conferred, and not their role as mediators of ADE -- a distinct context in which we agree with the Reviewer that a system in which expression of the human repertoire of FcR is desirable.

We have noted this limitation in the discussion section.

Lines 577 to 580: "Third, although anti-DENV antibodies can also have infection-enhancing properties, we do not measure this here since we focus on studying the neutralizing capacity of antibodies using cell substrates that do not express Fcγ receptors that can mediate antibody-dependent enhancement."

We also would like to point out that several published studies have addressed some of the questions mentioned by the Reviewer, using cells expressing different Fc receptors (Brandt *et al.*, 1982; Putnak *et al.*, 2008; Dejnirattisai *et al.*, 2016).

Brandt, W.E. *et al.* (1982) 'Infection enhancement of dengue type 2 virus in the U-937 human monocyte cell line by antibodies to flavivirus cross-reactive determinants', *Infect. Immun.*, 36(3), pp. 1036–1041.

Dejnirattisai, W. *et al.* (2016) 'Dengue virus sero-cross-reactivity drives antibody-dependent enhancement of infection with Zika virus', *Nat. Immunol.*, 17(9), pp. 1102–1108.

Putnak, J.R. *et al.* (2008) 'Comparative evaluation of three assays for measurement of dengue virus neutralizing antibodies', *Am. J. Trop. Med. Hyg.*, 79(1), pp. 115–122.

- *The threshold for specificity and sensitivity are not well described, whilst being important measures for assessing their data.*

R/ We agree with the Reviewer. We edited the Methods section and the Results section to improve clarity:

Lines 234 to 239: “Receiver Operator Curve (ROC) analysis was performed using the pROC package (v.1.18.0) in R studio (v. 2022.12.0+35). The pooled area under the curve (AUC), sensitivity and specificity to detect a subsequent symptomatic infection was estimated across different assay conditions. The cut-off of the assay was defined as the NT₅₀ value under which at least 90% of the subsequent symptomatic infections were detected (sensitivity). The specificity corresponded to the proportion of inapparent infections above this same value.”

Lines 453 to 457: “We performed a ROC analysis to compare the assay conditions in terms of their sensitivity and specificity (Table 3). The cut-off of the assay was defined as the NT₅₀ value under which at least 90% of the subsequent symptomatic infections were detected (sensitivity). The specificity corresponded to the proportion of inapparent infections above the cut-off value.”

To improve the interpretability of our results, we also included the following:

Lines 462 to 468: “For the assay performed on partially mature virions on standard Vero cells, which is the assay used in most clinical trials for vaccine testing, the NT₅₀ cut-off for achieving a sensitivity of at least 90% was 898. Nevertheless, at this titer, the specificity was only 8%. This means that, using this assay, although more than 90% of the individuals had an NT₅₀ below 898 before developing a symptomatic DENV infection, 92% of the individuals that subsequently developed an inapparent infection were also below this cut-off and yet were protected from disease.”

- In general the magnitude of the effects are very low. Given that the sample size is fairly low, it is slightly difficult to gauge whether these differences are biologically significant.

R/ Regarding the effect sizes, we added “Reviewer #3 Supplementary Table 1” (below) summarizing the observed effect sizes and power of our models for detecting protection from symptomatic disease for each incoming serotype and each assay condition. As noted in the table, many of the effect sizes for our models were considered medium according to f² classification by Cohen (effect size (f²) ≥ 0.02, small; ≥ 0.15, medium; ≥0.35, large) (Cohen, J., 2013), particularly for models of incoming DENV1 and partially for DENV2, whereas the effect size for incoming DENV3 was very small despite a relatively large sample size. Thus, this once again provides evidence of differences in the biological significance of NT₅₀ values by serotype.

Reviewer #3 Supplementary Table 1. Effect size and power estimation for the regression models of the association between NT₅₀ values under each assay condition with clinical outcome

	DENV1		DENV2		DENV3	
	Effect size (f2) ^a	Power	Effect size (f2)	Power	Effect size (f2)	Power
Mature Vero	0.30353	0.61965	0.19141	0.91501	0.00089	0.05452
Part. Mature Vero ^b	0.23690	0.51646	0.05772	0.44843	0.03430	0.23304
Mature Vero DC-SIGN	0.12804	0.31312	0.09877	0.66785	0.00606	0.08106
Part. Mature Vero DC-SIGN ^a	0.30237	0.61800	0.06445	0.48965	0.00991	0.10130

^a Effect size (f2) \geq 0.02 small, \geq 0.15 medium, \geq 0.35 large according to Cohen (Cohen, J., 2013).

^b Part. Mature = Partially mature virion.

In addition, the fact that we found different effect sizes depending on the incoming serotype highlights the differential relevance of NT₅₀ for conferring protection from subsequent DENV1, DENV2, and DENV3 infections. This analysis also explains why we did not detect a significant association between NT₅₀ magnitude and protection from symptomatic disease for an incoming DENV3 infection, at comparable sample sizes. Therefore, the differences in effect sizes reveal a relevant biological difference.

We have explained this in the Discussion section:

Lines 580 to 587: “Lastly, our current sample size is under-powered for the predictive model of risk of symptomatic disease for incoming DENV1 or DENV3 infection. Nonetheless, the observed differences in effect sizes imply that for achieving a power of at least 80% we would need only 28 pre-DENV1 infection individuals, in contrast with 231 pre-DENV3 infection individuals and a total of 222 individuals in the post-secondary infection group. These differences in the required sample size are evidence of the potential negligible role of nAbs in protection against DENV3 symptomatic infection and upon a post-secondary infection.”

- The choice of statistics are also not clear – it would be useful to explain why a paired Wilcoxon test was used in Fig 1.

R/ The Wilcoxon test is the non-parametric test that was used in Figure 1 given that our data does not follow a normal distribution. The paired version of the test was used because the variables we addressed are not independent of each other. To clarify this, we have added “Reviewer #3 Supplementary Figure 1” (below). In this Figure, it is possible to appreciate the distribution of the variables (in the diagonal of the correlation chart), the correlation coefficients, and the p-value of these correlations. Since all the variables are correlated among each other, they are not independent, and therefore the paired Wilcoxon test is the correct test for hypothesis testing.

Reviewer #3 Supplementary Figure 1. Correlation chart of the NT₅₀ assessed under different assay conditions. pm = partially mature, m=mature, V= Vero cells, VDC= Vero cells-DC SIGN. The upper triangle (red) shows the R² among variables and the statistical significance of the correlation (*p<0.05, **p<0.01, ***p<0.001). The lower triangle (blue) shows the correlation plots between variables.

- Similarly, Fig 6C shows a more pronounced difference in pre-inapp sample compared to pre-symptomatic ones, yet the difference is non-significant.

R/ The Reviewer is correct that in Fig 6C, although there is a more pronounced difference in pre-inapparent infection samples compared to pre-symptomatic samples, the difference remains non-significant. This is because non-parametric tests evaluate differences in the distribution of variables by ranks and not absolute values.

- Some figure legends need to be amended – e.g. Figure 5C is not described. Similarly, Fig 3B description needs to be clarified.

R/ Note that the original Figure 5 is now Figure 4. The figure legends were amended in response to the Reviewer’s comment as follows:

Lines 765 to 771: “Figure 3. Cross-reactive nAb titers of pre-inapparent and pre-symptomatic DENV infection samples. (A) NT₅₀ of pre-inapparent (n=64) and pre-symptomatic (n=63) DNV infection groups measured against the incoming serotype across FRNT assay conditions, and (B) NT₅₀ of pre-inapparent and pre-symptomatic participants stratified by their infection history at the time of sample collection (Post-1°: post-primary infection, Post-2°: post-secondary infection). Diamond indicates mean NT₅₀. Asterisks indicate Benjamini Hochberg adjusted p-values for Wilcoxon test. p-values: ns, >0.05; *, <0.05; **, <0.01; ***, <0.001; ****, <0.0001.”

Lines 772 to 777: “Figure 4. Comparison of nAb responses between pre-inapparent post-primary and post-secondary versus pre-symptomatic post-secondary infection samples. (A) Schematic of the groups compared. (B) Magnitude of the nAb response using different virion maturation states

and cell substrates. (C) Impact of change in assay condition on the neutralization titers of participants. Asterisks indicate Benjamini Hochberg adjusted p-values for Wilcoxon test (B) and paired Wilcoxon test (C). p-values: ns, >0.05; *, <0.05; **, <0.01; ***, <0.001; ****, <0.0001.”

- It would also be important to include assessment of cross-reactive antibodies between different serotypes (i.e post 1st DENV1 antibodies vs pre-inapparent DENV3 tested with DENV3- since the samples are already available).

R/ We agree with the reviewer that it is important to assess the level of cross-reactive antibodies between different serotypes. Indeed, all the measured nAbs in our study correspond to cross-reactive neutralizing antibodies. We have clarified this in the manuscript:

Lines 294 to 298: “In each case, NT₅₀ titers were measured against the incoming heterotypic serotype of each participant and compared using a paired Wilcoxon test. For instance, the NT₅₀ of serum samples from participants collected prior to a DENV1 infection was measured using DENV1, while the NT₅₀ of serum samples from participants collected prior to a DENV2 infection was measured using DENV2, etc.”

In our dataset, we did not identify homotypic infections among participants for whom serotyping was possible (i.e., post-primary infection); thus, all the assessed nAbs presumably correspond to cross-reactive neutralizing antibodies elicited against the incoming serotype, as suggested by the Reviewer. For instance, as part of a separate, ongoing study, we have characterized the infection history of the individuals who had a post-primary DENV3 infection. In this group, we identified 8 individuals with a primary DENV1 infection, 16 individuals with a primary DENV2 infection, and 4 individuals with a primary DENV4 infection. All these were tested in our current work to examine cross-reactive nAbs against the incoming serotype (DENV3), thus assessing the cross-reactive nAbs elicited. We acknowledge that it would be interesting to break down the differential impact of the composition of the nAb repertoire depending on the prior infecting serotype, but we are currently underpowered for such analysis, as there were not enough individuals with specific serotype sequence infection histories per group. We noted this as a limitation in the discussion section:

Lines 574 to 577: “Second, as our study population is from Nicaragua over a 15-year period, our analysis is naturally limited to the circulating serotypes and epidemiology of DENV in the timeframe and area of study. Further, we were not able to address the impact of different sequences of infections in the composition of the nAb response due to small sample sizes of specific serotype sequences.”

REVIEWERS' COMMENTS

Reviewer #3 (Remarks to the Author):

The authors have clarified all the concerns that I raised. I do not have any further comments.